# Towards an Evaluation Framework for Expressive Stream Reasoning

Pieter Bonte[0000−0002−8931−8343], Filip De Turck[0000−0003−4824−1199], and Femke Ongenae[0000−0003−2529−5477]

IDLab, Ghent University – imec
Pieter.Bonte@UGent.be

**Abstract.** Stream Reasoning, and more particular RDF Stream Processing (RSP), has focused on processing data streams in a timely manner, while expressive reasoning techniques, such as OWL2 DL, allow to fully model and interpret their domain knowledge. However, expressive reasoning techniques have thus far mostly focused on static data, as it tends to become slow with growing datasets. Expressive Stream Reasoning aims to combine these fields and evaluate expressive reasoning techniques in a timely manner over volatile data streams through various reasoning optimizations. Both expressive reasoning and RSP have benchmarks and frameworks for evaluating and comparing proposed solutions. However, no benchmarks or evaluation frameworks for Expressive Stream Reasoning are currently available.

In this paper, we propose *OWL2Streams*, a resource framework for the evaluation of Expressive Stream Reasoning solutions. We identified challenges and opportunities for optimizations when dealing with expressive reasoning over the combination of streams and static data. *OWL2Streams* proposes three streaming scenarios, each tackling different challenges.

**Keywords:** Stream Reasoning · Evaluation · Benchmark

## 1   Introduction

Due to the increasing rate at which data is being produced, the Stream Reasoning community has investigated how to process heterogeneous data streams in a timely manner. RDF Stream Processing (RSP), a part of the Stream Reasoning initiative, is focusing specifically on the timely processing of RDF data streams.

However, to extract meaningful insights from these data streams, the streams should be combined with domain knowledge [2, 5]. Expressive reasoning, such as OWL2 DL, allows to fully capture the domain knowledge. However, until now, such reasoning techniques have mostly focussed on rather static data [10]. Expressive stream reasoning aims at optimizing and exploiting trade-offs to speed up the slower expressive reasoning techniques, such that they can be used for streaming scenarios [3]. However, currently, there are no benchmarks or evaluation frameworks available to evaluate and push the research field forward. The RSP community has investigated multiple benchmarks for low expressive streaming tasks [1], while the reasoning community has proposed benchmarks

and evaluation frameworks for highly expressive static tasks [9]. Currently, a benchmark or evaluation framework that bridges the two is still missing.

In this paper, we propose *OWL2Streams*, the first initiative towards an expressive reasoning evaluation framework resource. Similar to the OWL2Bench benchmark [9], we provide challenges for each of the OWL2 profiles. Furthermore, we highlight additional challenges when performing expressive reasoning over volatile data. Each of the challenges can be used to optimize different parts of the expressive stream reasoning process. We propose three use cases, i.e. a Smart City, Smart Building, and a University Management Scenario, each tackling a different combination of challenges. Next to the streaming data, each scenario provides an ontology TBox and an ABox describing the static data. *OWL2Streams* builds upon prominent Semantic Web resources, such as RML [6], TripleWave [7] and VoCaLS [12].

## 2   Framework setup

For the setup of the *OWL2Streams* evaluation framework, we use TripleWave [7] for the publishing of the streams and VoCaLS [12] for the stream description. We use the idea of a *Test Stand* [11], where the framework controls the input and validates the output of the engine under evaluation. *OWL2Streams* consists of the following building blocks:

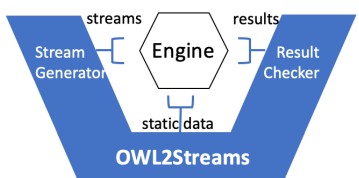

**Fig. 1.** The OWL2Streams test stand

- The *Stream Generator* generates data streams and makes them accessible to the engine under evaluation.
- The *Result Checker* functions as an *oracle*, that has an overview of which results to expect. It joins forces with the *Stream Generator* to realize this.
- The framework also provides the *static data*, which needs to be taken into account during the evaluation.

*OWL2Streams* is publicly available on Github[1].

## 3   Expressive Stream Reasoning Challenges

When performing expressive reasoning over the combination of both static and streaming data, there are additional challenges to tackle, fortunately each challenge also contains opportunities for further optimizations:

- *Size of the static data*: the larger the size of the static data, the slower the reasoning. Optimizations can focus on only extracting relevant parts of the static data [4].
- *Expressivity of the ontology*: the expressivity negatively impacts the reasoning time, optimizations can focus on extracting only relevant statements from the ontology TBox.

---

[1] https://github.com/IBCNServices/OWL2Streams

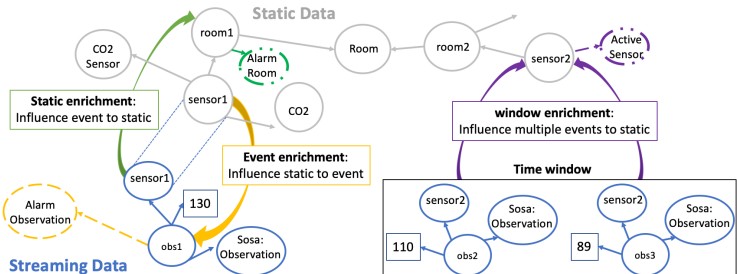

**Fig. 2.** Different levels of inference influences between static and streaming data.

– *Frequency of the stream*: the faster the updates in the stream, the faster the reasoning process needs to be able to provide results. Optimizations can focus on first filtering relevant facts from the data [3].
– *Number of parallel streams*: More streams imply more data, however, challenges arise when data from various streams need to be correlated.
– *Numeric data streams*: in domains such as the IoT, many streams contain numeric data. However, data property reasoning is not supported by many OWL2 reasoners. Optimizations could focus on filtering out events with numeric values that are not useful.
– *Data Expiration*: next to reasoning over the addition of data, the removal of data is even more complex. Optimizations could investigate adapted incremental reasoning techniques for expressive reasoning. (Currently, none exist for OWL2 DL.)

The inference influence between the static and streaming data imposes different challenges as well. We identify four influence cases, as visualized in Figure 2:

– *Static enrichment:* the influence of events in the data stream on the static data. The addition of the events causes additional inferences on the static data, e.g., inference that a *Room* is an *AlarmRoom*.
– *Event enrichment:* the influence of the static data on events. The enrichment of the events with static data allows to infer additional facts about the events.
– *Window enrichment:* the influence of multiple events on the static data. To infer additional facts about the static data, multiple events from the streams need to be considered. Therefore, events cannot be processed one at a time and need to be combined together in a time window.

We make these distinctions, as it allows optimization in each area.

## 4   Streaming Scenarios

*OWL2Streams* currently supports three scenarios, each focusing on different challenges presented above. The challenges tackled by each scenario are presented in Table 1. Note that for some scenarios, the size of the static data, or the number of streams can be generated and can thus vary in size.

$S_1$ An adaptation of the OWL2Bench benchmark [9]. This scenario consists of a university where students register and enroll in a certain program. The data stream consists of the enrollment of the students.

S$_2$ An extension of the CityBench benchmark [1], containing more elaborate background knowledge. This scenario describes a typical Smart City case.

S$_3$ A new Smart building COVID-19 scenario, consisting of sensors that measure the air quality in various rooms. The domain knowledge allows to infer if certain rooms have higher probabilities for COVID-19 infections, based on the air quality and the activities in each room [8].

**Table 1.** Mapping of the various challenges on the three Scenarios. (D+: discretization)

| | Static Size | Expressivity | Stream Frequency | Number parallel streams | Numeric Streams | Data Expiration | Inference Influence |
|---|---|---|---|---|---|---|---|
| **S1** | Small to very large | High | Slow | 1 | No | No | Static & Event |
| **S2** | Small to medium | Medium | High | 1 - 10 | Yes & D+ | Yes | All |
| **S3** | Small | Low | High | 1 - 100 | Yes | Yes | All |

## 5   Conclusion

In this paper, we present *OWL2Streams*, a first evaluation framework for expressive stream reasoning. We have identified various challenges that arrise when performing expressive reasoning over volatile data streams. *OWL2Streams* consist of three scenarios that each tackle a different combination of challenges.

**Acknowledgments:** Pieter Bonte is funded by a postdoctoral fellowship of Fonds Wetenschappelijk Onderzoek Vlaanderen (FWO) (1266521N).

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
