# OpenReview forum: "Towards an Evaluation Framework for  Expressive Stream Reasoning"
_eswc-conferences.org/ESWC/2021/Conference/Poster_and_Demo_Track — ESWC2021 P&D_

### Official Review · AnonReviewer4 · 2021-04-11
**Review of Paper18**

**Rating:** 6
**Confidence:** 3

**Review:**

Quality: good

Clarity: fair

Originality: good

Significance of this work: good

Pros:
1.	his paper proposes one resource framework called OWL2Stream for evaluating the expressive stream reasoning solutions for three streaming scenarios.
2.	This work concludes the challenges and opportunities for optimizations for dealing with expressive reasoning.
3.	The source code can be assessed, and the authors give detailed information for its reproducibility.

Cons:
1.	The content of figure 2 is hard to understand by its related statements.
2.	Lack some explanation of the discrete results for each challenge (e.g., inference influence) in three scenarios.
3.	It is hard to see the concrete improvement for this framework in view of quantification.


**Anonymity:**

Yes, I would like my review to remain anonymous.

---

### Official Review · AnonReviewer2 · 2021-04-16
**Expressivity - selling point of the paper but with little substantiation**

**Rating:** 4
**Confidence:** 4

**Review:**

The authors present preliminary work for an evaluation framework for stream reasoning systems, where the title highlights the focus on expressivity.

Regarding the importance of ontology expressivity in the approach and highlighted in the submission title, I would hope that expressivity is a key criterion and not just one among nine. Moreover, the submission just says that reasoning negatively impacts the execution time. I guess there would be more to say about expressivity:  What kind of expressivity do you target? Monotonous reasoning? Over what time window do you intend to evaluate negation? What is the relation to modal and temporal logics?

Regarding the other criteria of the call for submissions:
* The work is indeed relevant to the Semantic Web
* In terms of originality, I would say to have an evaluation framework would be a natural extension of the works in the stream reasoning Semantic Web sub-community.
* In terms of topicality, I think that the stream reasoning sub-community is currently maturing so the submission is timely
* In terms of potential significance, I would have hoped for a stronger motivation why stream reasoning and why in particular stream reasoning with ontologies on the high end of the expressivity spectrum.


**Anonymity:**

Yes, I would like my review to remain anonymous.

---

### Official Review · AnonReviewer1 · 2021-04-16
**hope to see it as a resource paper in the future**

**Rating:** 6
**Confidence:** 3

**Review:**

# content of the poster
The poster describes a expressive reasoning benchmark that combines and extends existing benchmarks for streaming and static data. It focuses on multiple domains and different challenges.

# relevance to the semantic web
the proposed benchmark is certainly relevant to the semantic web, particularly to the stream processing. no issues here.

# potential impact

The work seems to have a significant potential impact as such benchmarks are available rather limitedly.  It also has multiple domains with different characteristics in terms of reasoning challenges, which is a neat feature. However, there is no future work described, which is I think an important part of a poster paper. Will the competency of the benchmark be evaluated? Is it going to be used in some academic or industrial projects?

# topicality
RDF Stream Processing, including reasoning has been increasingly important within the last 10 - 15 years, given the developments in IoT, wearable computers etc. The work has no issues with regards to topicality.

# clarity
 The paper is well-written, no major issues with clarity. Some small things I would like to point out:
- "Room is an AlarmRoom" sounds like there is a subsumption hierachy between Room and AlarmRoom. Maybe this is the case, but to me AlarmRoom sounds like a more specific type than Room.
- in abstract more particular -> more particularly
- it is better to explain with one sentence, or a footnote what RML, VoCaLS and TripleWave are.

# originality
It is true that the RSP benchmarks more focus on querying and less expressive reasoning tasks. The expressive reasoning task on streaming data is not widely covered, so overall the paper touches a *relatively* uncovered part of the RSP. However, I think the authors should briefly mention how does it compare to LASS[1]? It is a highly relevant work as it is also proposing a benchmark for the same task.
It would be also relevant to look into the LarKC [2] project for some context and motivation:

# summary
I think the work is suitable for a poster paper. However,  the authors should focus a bit more on the future work and potential impact, and take a look again strongly relevant previous work.

[1] Tommasini, R., Balduini, M., & Valle, E.D. (2017). Towards a Benchmark for Expressive Stream Reasoning. RSP+QuWeDa@ESWC 2017. http://ceur-ws.org/Vol-1870/paper-03.pdf

[2] https://www.larkc.org/resources/index.html

**Anonymity:**

Yes, I would like my review to remain anonymous.

---

### Official Review · Program_Chairs · 2021-04-18
**Metareview: Accept (But focus on expressivity requires clarification)**

**Rating:** 6
**Confidence:** 5

**Review:**

This was a borderline paper with two reviewers learning towards an accept, and one leaning towards a reject. In terms of the more negative review, we believe that the motivation for exploring more expressive forms of reasoning could be clarified for the final version. Hence we think that the paper can be accepted and encourage the authors to carefully consider the reviewers' feedback for the final version.

**Anonymity:**

Yes, I would like my review to remain anonymous.

---

### Decision · Program_Chairs · 2021-04-19

Accept